# High-Performance Surface-Enhanced Raman Scattering Substrates Based on the ZnO/Ag Core-Satellite Nanostructures

**DOI:** 10.3390/nano12081286

**Published:** 2022-04-10

**Authors:** Qianqian Sun, Yujie Xu, Zhicheng Gao, Hang Zhou, Qian Zhang, Ruichong Xu, Chao Zhang, Haizi Yao, Mei Liu

**Affiliations:** 1Collaborative Innovation Center of Light Manipulations and Applications in Universities of Shandong, School of Physics and Electronics, Shandong Normal University, Jinan 250014, China; 2019020497@stu.sdnu.edu.cn (Y.X.); 201809020304@stu.sdnu.edu.cn (Z.G.); 2020020573@stu.sdnu.edu.cn (H.Z.); 202009100522@stu.sdnu.edu.cn (Q.Z.); 202009100431@stu.sdnu.edu.cn (R.X.); chaozhang@sdnu.edu.cn (C.Z.); 2Key Laboratory of Smart Lighting in Henan Province, School of Energy Engineering, Huanghuai University, Zhumadian 463000, China

**Keywords:** SERS, hierarchical hybrid structures, semiconductor micro/nanostructures, noble metal nanoparticles, metal oxide

## Abstract

Recently, hierarchical hybrid structures based on the combination of semiconductor micro/nanostructures and noble metal nanoparticles have become a hot research topic in the area of surface-enhanced Raman scattering (SERS). In this work, two core-satellite nanostructures of metal oxide/metal nanoparticles were successfully introduced into SERS substrates, assembling monodispersed small silver nanoparticles (Ag NPs) on large polydispersed ZnO nanospheres (p-ZnO NSs) or monodispersed ZnO nanospheres (m-ZnO NSs) core. The p-ZnO NSs and m-ZnO NSs were synthesized by the pyrolysis method without any template. The Ag NPs were prepared by the thermal evaporation method without any annealing process. An ultralow limit of detection (LOD) of 1 × 10^−13^ M was achieved in the two core-satellite nanostructures with Rhodamine 6G (R6G) as the probe molecule. Compared with the silicon (Si)/Ag NPs substrate, the two core-satellite nanostructures of Si/p-ZnO NSs/Ag NPs and Si/m-ZnO NSs/Ag NPs substrates have higher enhancement factors (EF) of 2.6 × 10^8^ and 2.5 × 10^8^ for R6G as the probe molecule due to the enhanced electromagnetic field. The two core-satellite nanostructures have great application potential in the low-cost massive production of large-area SERS substrates due to their excellent SERS effect and simple preparation process without any template.

## 1. Introduction

Surface-enhanced Raman Scattering (SERS) as a powerful spectroscopy technique has been widely used in chemistry, pharmacy, food safety, and environmental monitoring field due to its high sensitivity, fast response, and non-destructive data collection [1,2,3,4,5,6]. Due to the strong localized surface plasmon resonance (LSPR), noble metal nanoparticles are widely applied in SERS substrate [7,8,9,10,11]. Noble metal nanoparticles can be produced by chemical synthesis, chemical etching, electrochemical deposition, sputtering, thermal evaporation method, nanosphere lithography, electron beam lithography, and laser annealing [12,13,14]. Noble metal colloidal nanoparticles produced by the chemical synthesis method are prone to aggregate, which will reduce the activity and stability of SERS substrates. The noble metal colloidal nanoparticles by other methods are hard to make highly uniform and reproducible SERS signals due to the existence of random hot spots [15,16]. The hierarchical hybrid SERS substrates composed of semiconductor micro/nanostructures and noble metal nanoparticles are research highlights to address this issue [17,18,19,20]. The semiconductor micro/nanostructures can provide a larger surface area and better support structures for the deposition of metal nanoparticles, which can increase the density and intensity of hot spots. The different hierarchical hybrid SERS substrates have been constructed based on the different semiconductors, such as Zinc oxide (ZnO), aluminum oxide (AAO), cupric oxide (CuO), and titanium dioxide (TiO_2_) [21,22,23,24,25,26]. Among these semiconductors, the application of ZnO in SERS substrates is particularly prominent due to the advantages of various preparation methods, high chemical stability, high refractive index nature, abundant anisotropic nanostructures, adequate sources, and popular price [27,28,29]. Various ZnO micro/nanostructures have been employed in SERS substrates to improve Raman performance, such as nanosheet, nanorod, and nanocone [30,31,32,33]. However, most of the ZnO micro/nanostructures in recent typical works have been synthesized with the help of the template, as shown in Table 1. This property may limit the application of ZnO micro/nanostructures in the low-cost massive production of SERS substrates. Therefore, the massive production of hierarchical hybrid SERS substrates based on semiconductor/noble metals is still a big challenge in a simple and low-cost production way.

In this work, the two core-satellite nanostructures of metal oxide/metal nanoparticles were successfully introduced into the SERS substrates assembling monodispersed small silver nanoparticles (Ag NPs) on large polydispersed ZnO nanospheres (p-ZnO NSs) or monodispersed ZnO nanospheres (m-ZnO NSs) core. The p-ZnO NSs and m-ZnO NSs were synthesized by the pyrolysis method without any template. The schematic diagram for the fabrication process of p-ZnO NSs and m-ZnO NSs is shown in Figure 1. The p-ZnO NSs methanol solution and m-ZnO NSs methanol solution were dripped onto the cleaned silicon (Si) substrates, respectively. The Ag NPs were prepared by the thermal evaporation method without any annealing process, as shown in the Appendix A. The ultralow limit of detection (LOD) of 1 × 10^−13^ M is achieved in the two core-satellite nanostructures with Rhodamine 6G (R6G) as the probe molecule. Compared with the Si/Ag NPs substrate, the two core-satellite nanostructures of Si/p-ZnO NSs/Ag NPs and Si/m-ZnO NSs/Ag NPs SERS substrates have higher enhancement factors (EF) of 2.6 × 10^8^ and 2.5 × 10^8^ for R6G as the probe molecule due to the enhanced electromagnetic field. The two core-satellite nanostructures of p-ZnO NSs/Ag NPs and m-ZnO NSs/Ag NPs have great application potential in the low-cost massive production of large-area SERS substrates due to their excellent SERS effect and simple preparation process without any template.

## 2. Results

As schematically displayed in Figure 1, the p-ZnO NSs and m-ZnO NSs were prepared by the pyrolysis method with a two-stage reaction process. Microwave heating may promote the hydrolysis of Zn(OAc)_2_·2H_2_O in diethylene glycol (DEG). The reactions may occur according to the following two equations [34]. The Zn-complex would be formed by the hydrolysis reaction for Zn(OAc)_2_·2H_2_O according to Equation (1). The pure ZnO nanocrystal would be formed by the dehydration and acetic acid removal of the Zn-complex during the aging time, as shown in Equation (2). These fresh-formed ZnO nanocrystals spontaneously aggregate to form ZnO nanoparticle clusters in the role of intermolecular forces [35].
(1)$$\mathrm{Zn}({\mathrm{CH}}_{3}\mathrm{COO}{)}_{2}+{\mathrm{xH}}_{2}O\;\overset{\Delta }{\to }\;\mathrm{Zn}(\mathrm{OH}-)x\;{\left({\mathrm{CH}}_{3}\mathrm{COO}-\right)}_{2-x}+{\mathrm{xCH}}_{3}\mathrm{COOH}$$
(2)$$\mathrm{Zn}(\mathrm{OH}-)x\;{\left({\mathrm{CH}}_{3}\mathrm{COO}-\right)}_{2-x}\;\overset{\Delta }{\to }\;\mathrm{ZnO}+(x-1)H_{2}O+(2-x){\mathrm{CH}}_{3}\mathrm{COOH}$$

The size of ZnO nanoparticle clusters can be tuned by simply adjusting the amount of Zn-complex precursors while keeping all other parameters constant. The varying additions of Zn-complex solution may arise slight differences in the amount of H_2_O and crystal nuclei. Both higher H_2_O content and higher Zn-complex concentration could accelerate the hydrolysis of Zn(OAc)_2_ and dehydration of the newly formed Zn-complex, thus leading to more nuclei in the bulk solution and finally smaller ZnO nanoparticle clusters [35]. The higher Zn-complex concentration in the synthesis of m-ZnO NSs may lead to more nuclei in the bulk solution and finally smaller ZnO nanoparticle clusters with smaller diameters. To verify the quality of the p-ZnO NSs and m-ZnO NSs, the X-ray diffraction (XRD) spectra and the X-ray photoelectron spectroscopy (XPS) spectra were measured as shown in Figure 2a–c. The (100), (002), (101), and (102) diffraction peaks of the wurtzite ZnO structure (JCPDS card No. 36–1451) can be seen in the two XRD spectra, which means that the p-ZnO NSs and the m-ZnO NSs both are crystalline. The stronger and sharper peaks of the m-ZnO NSs mean that m-ZnO NSs have higher crystallinity compared with the p-ZnO NSs. The grain sizes of ZnO nanocrystals are approximately estimated to be 9 nm and 14 nm for p-ZnO NSs and m-ZnO NSs by using the Scherrer equation from the full width at half maximum (FWHM) of diffraction. Both XPS spectra of Zn 2p for the p-ZnO NSs and m-ZnO NSs present two independent symmetrical peaks with binding energies at 1022 and 1045 eV. The two independent symmetrical peaks have been attributed to Zn 2p_3/2_ and Zn 2p_1/2_, respectively. The binding energy difference between the two peaks is 23 eV, which is consistent with the characteristic value of ZnO. The data show that Zn in the p-ZnO NSs and m-ZnO NSs mainly exist in the Zn^2+^ form [36,37]. The XPS spectrum of O 1s for the p-ZnO NSs presents a symmetrical peak with binding energies at 532 eV, which is attributed to the lattice oxygen of O^2−^ (Zn-O) in p-ZnO NSs. The XPS spectrum of O 1s for the m-ZnO NSs presents two non-independent shoulder peaks, which can be fitted into two Gaussian peaks at 532 eV and 530 eV. The peaks of 532 eV and 530 eV could be separately attributed to the lattice oxygen of O^2−^ (Zn-O) in m-ZnO NSs and -OH (Zn-OH) absorbed onto the surface of the ZnO NSs due to structural defects [37,38]. The data show that p-ZnO NSs have higher quality due to no structural defects compared with the m-ZnO NSs.

The morphologies of the synthesized p-ZnO NSs at the first stage were obtained by scanning electron microscopy (SEM), as shown in Figure 2d,e. The statistical histograms for the diameters of the p-ZnO NSs are shown in Figure 2f. The p-ZnO NSs consists of ZnO nanospheres (ZnO NSs) with different diameter distribution ranging from 50 to 420 nm, and the average diameter of p-ZnO NSs is about 200 nm. A series of the large ZnO NSs (≥200 nm) surrounded by a ring of small ZnO NSs (<200 nm) can be regarded as a major component of the p-ZnO NSs. The average diameter of large ZnO NSs and small ZnO NSs in p-ZnO NSs are 276 nm and 135 nm, respectively. The schematic diagram of the p-ZnO NSs is shown in Figure 2g. The morphologies of the synthesized m-ZnO NSs at the two stages were measured by SEM, as shown in Figure 2h,i. The statistical histograms for the diameters of the m-ZnO NSs are shown in Figure 2j. The m-ZnO NSs consist of homogeneous oval-shaped ZnO NSs with long diameter distribution ranging from 108 to 223 nm and short diameter distribution ranging from 91 to 201 nm. An oval-shaped ZnO NSs can be surrounded by a ring of other oval-shaped ZnO NSs in the m-ZnO NSs. The average long diameter and short diameter for oval-shaped ZnO NSs are 162 nm and 132 nm, respectively. The schematic diagram of the m-ZnO NSs is shown in Figure 2k. In Figure 2f,j, the statistical diameters of the p-ZnO NSs and the m-ZnO NSs are fitted with Gaussian distribution. The width of the particle size distribution is described by the polydispersity index (Pdi). The relative polydispersity can be calculated by % Polydispersity (%Pd) = Coefficient of variation = (PDI)^1/2^ × 100. The samples with %Pd < 20% can be seen as monodisperse as a rule of thumb. The detailed calculation method of %Pd can be found in the Appendix A. The calculated %Pd of diameter for p-ZnO NSs, long diameter, and short diameter for the m-ZnO NSs are 45.5%, 15.2%, and 16.7%, respectively. Thus, the definitions of p-ZnO NSs and m-ZnO NSs in this work are correct. From the SEM images with lower magnification (Figure 2d,h), the p-ZnO NSs and the m-ZnO NSs can both effectively cover the substrates with large areas, which means that they can be applied to the large area SERS substrates. From the magnified SEM images of p-ZnO NSs and m-ZnO NSs (Appendix A), the surface of ZnO NSs is rough, which reveals that a single ZnO NSs with a rough surface is formed by the aggregate of an amount of ZnO nanocrystals [39]. The rough surface of ZnO NSs is more favorable for the formation of regular and orderly Ag NPs on the ZnO NSs surface to enhance SERS activity [40].

The deposition thickness of Ag by thermal evaporation does have an important influence on the size and morphology of the Ag NPs on the ZnO NSs. The SEM images of p-ZnO NSs/Ag NPs and m-ZnO NSs/Ag NPs with different deposition thicknesses (3 nm, 6 nm, 9 nm) of Ag by thermal evaporation are shown in Appendix A. Compared with the Ag NPs on the p-ZnO NSs or m-ZnO NSs with deposition thickness of 3 nm and 9 nm, the Ag NPs on the p-ZnO NSs or m-ZnO NSs with deposition thickness of 6 nm has a stronger sense of particles. Too small deposition thickness may make the small scale of Ag NPs form obvious Ag NPs. Too large deposition thickness is easy to make too large Ag NPs and tend to form continuous films. The localized LSPR effect of Ag NPs in the hot spots is exist at the nanogaps between two Ag NPs or the apex of metal tips. Thus, the Ag NPs with a deposition thickness of 6 nm are deposited on the p-ZnO NSs and m-ZnO NSs to prepare the SERS substrate. The two core-satellite nanostructures are successfully prepared as shown in the SEM images of p-ZnO NSs/Ag NPs (Figure 3a,b) and m-ZnO NSs/Ag NPs (Figure 3e,f). Relatively regular and orderly Ag NPs are adsorbed on the surface of the p-ZnO NSs or m-ZnO NSs due to the interaction between molecules. Relatively regular and orderly Ag NPs look like satellites around the core of zinc p-ZnO NSs or m-ZnO NSs. The statistical histograms for the diameters of the Ag NPs and gaps between adjacent Ag NPs on the surface of p-ZnO NSs and m-ZnO NSs are shown in Figure 3c,g. The average diameter of the Ag NPs and the average gap between adjacent Ag NPs on the surface of p-ZnO NSs are about 32 nm and 3 nm, respectively. The average diameter of the Ag NPs and the average gap between adjacent Ag NPs on the surface of m-ZnO NSs are about 25 nm and 10 nm, respectively. The schematic diagrams of the p-ZnO NSs/Ag NPs and m-ZnO NSs/Ag NPs are shown in Figure 3d,h. These results suggest that the diameter of the Ag NPs and the average gap between adjacent Ag NPs are different at the same substrates with the same deposition thicknesses of Ag. The average diameter of nano spherical p-ZnO NSs is larger than that of oval-shaped m-ZnO NSs, which shows that the p-ZnO NSs have smaller curvature. The smaller curvature may be more favorable for the growth of Ag NPs, which may be one of the influencing factors for the larger average diameter of the Ag NPs and smaller the average gap between adjacent Ag NPs on the surface of p-ZnO NSs. The influencing factors for the formation process of Ag NPs on the ZnO templates are very complex, such as the shape, size, roughness, crystallinity of ZnO, and so on. In contrast, Ag NPs with the same deposition thicknesses of Ag were deposited on the Si substrate by thermal evaporation. The SEM images, statistical histograms for the diameters, and schematic diagram of the Ag NPs on the surface of the Si substrate are shown in Figure 4. The average diameter of the Ag NPs and the average gap between adjacent Ag NPs on the surface of the Si substrate are about 36 nm and 8 nm, respectively. The shape and number of Ag NPs on the surface of p-ZnO NSs or m-ZnO NSs become more spherical and more numerous compared with that on the surface of the Si substrate. It is worth noting that all Ag NPs were prepared without any annealing operation. It means that the p-ZnO NSs and m-ZnO NSs provide supportive structures for the deposition of Ag NPs on them, which play a key role in controlling the size and shape of Ag NPs [41]. The structured surfaces of p-ZnO NSs and m-ZnO NSs can provide large structure areas for the deposition of Ag NPs, which is beneficial to increasing the number and surface area of Ag NPs [42].

The Ag NP, p-ZnO NS, m-ZnO NS, p-ZnO NS/Ag NP, and p-ZnO NS/Ag NP structures were prepared on quartz glass substrates to test the absorption spectra. The absorption spectra of the different structures on quartz glass were measured by using a PERSEE TU-1900 spectrophotometer with a wavelength range of 300 nm−900 nm. The absorption spectra of the different structures on quartz glass are shown in Appendix A. The Ag NPs, p-ZnO NSs/Ag NPs, and m-ZnO NSs/Ag NPs show a wide absorption range of 300 nm–900 nm. The strong absorption peak of 543 nm is shown in Ag NPs and p-ZnO NSs/Ag NPs, which is the absorption peak of Ag NPs [43]. The m-ZnO NSs/Ag NPs have a strong absorption peak of 420 nm, which corresponds to the m-ZnO NSs itself absorption peak. The excitation wavelength of 532 nm was chosen for the SERS measurements because it is closer to the absorption peak of the Ag NPs used in this work. A series of Raman spectra were detected to evaluate the Raman enhancement ability of the two core-satellite nanostructures as SERS substrates under the same conditions. The concentration varying of R6G as the probe molecule is from 10^−7^ to 10^−13^ M. The initial Raman spectra have fluorescence from R6G since it absorbs at the wavelength of the used laser. The fluorescence backgrounds from R6G are removed to obtain clearer Raman signals and better analyze the role of the substrate. Initial Raman spectra and Raman spectra after removing the fluorescence background are shown in Appendix A for R6G with a concentration of 10^−7^ M on the Si/p-ZnO NSs/Ag NPs substrates. The ten Raman spectra were collected from ten batches for R6G with each concentration on the different substrates. The intensity of the SERS signal of R6G with each concentration is similar for different peaks. The Raman spectra given in this work are the average of 10 Raman spectra. Characteristic peaks of 613, 774, 1185, 1315, 1365, 1508, and 1650 cm^−1^ can be seen in the Raman spectra for R6G with different concentrations, as shown in Figure 5a,d. The vibrational modes for the characteristic peaks are listed in Appendix A. The typical Raman spectrum of R6G is still seen for R6G with a concentration of 10^−13^ M in Si/p-ZnO NSs/Ag NPs and Si/m-ZnO NSs/Ag NPs substrates. The SERS signal intensities are gradually increased along with the increase in concentrations for the R6G molecular. The EF of Si/p-ZnO NSs/Ag NPs and Si/m-ZnO NSs/Ag NPs substrates were calculated according to the equation:$$\mathrm{EF}=\frac{I_{SERS}/N_{SERS}}{I_{RS}/N_{RS}}$$
where *I_SERS_* and *I_RS_* are the intensity of the SERS and the normal Raman scattering under identical test conditions, respectively. *N_SERS_* and *N_RS_* are the number of probe molecules within the laser spot for SERS and normal Raman intensities experiments, respectively. Due to the same laser spot (1 μm), the value of *N_RS_*/*N_SERS_* was roughly calculated according to the ratio of the average areal density (AD) for R6G on flat Si substrate and SERS substrates. The detailed calculation process of the average AD can be seen in the Appendix A. The ADs of the R6G with a concentration of 10^−13^ M on the fabricated Si/p-ZnO NSs/Ag NPs and Si/m-ZnO NSs/Ag NPs substrates are around 3.0 × 10^−2^ and 2.4 × 10^−2^ molecules/μm^2^, respectively. A larger AD means better adhesion properties of the surface of the Si/p-ZnO NSs/Ag NPs substrate for the R6G. The adhesion properties can be analyzed according to the water contact angle. The water contact angles of the Si/p-ZnO NSs/Ag NPs and Si/m-ZnO NSs/Ag NPs substrates are 50° and 30°, as shown in Appendix A. The surface of Si/p-ZnO NSs/Ag NPs with higher contact angle exhibits greater adhesion of water droplets to the surface, which can increase the areal density of analytes on the substrate and thus improve the LOD and the SERS effect [44,45]. The Raman intensities of the 613 cm^−1^ peak of R6G with 10^−13^ M concentration are ~167 and 123 for Si/p-ZnO NSs/Ag NPs and Si/m-ZnO NSs/Ag NPs substrates, respectively. The flat Si substrate was selected as the reference substrate. The Raman spectra of R6G with 10^−2^–10^−5^ M concentrations on the flat Si substrates were detected and are shown in Appendix A. The typical Raman spectrum of R6G can be still seen at 10^−4^ M concentration for R6G in flat Si substrates, which come from the interference effect due to the multiple reflections of the incident laser [46,47]. The Raman intensity at 613 cm^−1^ peak of R6G with 10^−4^ M concentration is 416 for the flat Si substrate. The calculated EFs of Si/p-ZnO NSs/Ag NPs and Si/m-ZnO NSs/Ag NPs substrates are 2.6 × 10^8^ and 2.5 × 10^8^, respectively. These results indicate that the Si/p-ZnO NSs/Ag NPs and Si/m-ZnO NSs/Ag NPs as SERS substrates have high sensitivities.

To investigate the quantitative detection ability, the linear relationship between the Raman intensities at 613 cm^−1^ or 774 cm^−1^ peaks with the R6G molecular concentrations was fitted and is shown in Figure 5b,e. The error bars represent the intensity distribution of the 613 cm^−1^ and 774 cm^−1^ peaks from 10 spots on one sample. All error values are small compared with the corresponding Raman intensities, which shows good reproducibility of different samples. The high determination coefficients (R^2^) of 0.995–0.998 for 613 cm^−1^ and 774 cm^−1^ peaks indicate the good linear relationship of Raman intensities with different R6G concentrations. Meanwhile, the covering of the R6G on the surface of SERS substrates can be analyzed with the fitting of adsorption Langmuir isotherm between Raman intensities and the logarithm of R6G concentrations. The relationships between Raman intensities at 613 cm^−1^ and 713 cm^−1^ peaks and the logarithm of R6G concentrations (10^−7^–10^−13^ M) were fitted according to the Langmuir isotherm model, which is shown in Appendix A. The R^2^ for the Langmuir isotherm fitting of Si/p-ZnO NSs/Ag NPs substrates are 0.984 and 0.951 for 613 cm^−1^ peaks and 774 cm^−1^. The R^2^ for the Langmuir isotherm fitting of Si/m-ZnO NSs/Ag NPs substrates are 0.959 and 0.955 for 613 cm^−1^ peaks and 774 cm^−1^. The non-linear Langmuir isotherm fitting may be more suitable for the actual measurement of wide range concentrations.

In addition to sensitivity, homogeneity and reproducibility play an important role in practical applications of the SERS substrates. The ten Raman spectra of R6G with 10^−7^ M concentration were collected on the Si/p-ZnO NSs/Ag NPs and Si/m-ZnO NSs/Ag NPs substrates from different batches, as shown in Figure 5c and Appendix A. The relative intensities and the average value at the 613 cm^−1^ peaks of the ten Raman spectra are shown in Figure 5f. Minor fluctuations around the average intensity are exhibited in all intensities of the 613 cm^−1^ peak. The corresponding relative standard deviation (RSD) value are about 8.51% and 10.47% for the Si/p-ZnO NSs/Ag NPs and Si/m-ZnO NSs/Ag NPs substrates. These data demonstrate that the two core-satellite nanostructures can realize the homogeneous Raman signal and excellent reproducibility. The detection of malachite green (MG) molecule in aquatic products is a very important link. To investigate the potential of the Si/p-ZnO NSs/Ag NPs substrates in practical application, the Raman spectra of MG as probe molecule with different concentrations (10^−5^–10^−10^ M) were measured and are shown in Appendix A. The typical Raman peaks (804, 912, 1179, 1372, and 1625 cm^−1^) of the MG molecule are detected at high concentrations. The Raman intensities continued to weaken along with the decrease of the concentrations for the MG molecule. The main characteristic peaks of the MG molecule could be also observed at 10^−10^ M concentration, which further indicates that the Si/p-ZnO NSs/Ag NPs substrates as SERS substrates have a high sensitivity. The linear fit calibration curve (R^2^ = 0.981) is illustrated in Appendix A with error bars. For the MG molecule, the logarithms of the SERS intensities and the concentrations are proportional, which indicates the promising application prospects of this substrate.

To investigate the SERS effect of the two core-satellite nanostructures, some contrast substrates were prepared with different structures. The Raman spectra of contrast substrates are shown in Figure 6a for R6G with concentrations of 10^−7^ M, 10^−4^ M, or 10^−2^ M. The corresponding intensities of 613 cm^−1^ peak and 774 cm^−1^ peaks for different substrates are shown in Figure 6b. The weak Raman intensities are achieved for the Si/p-ZnO NSs or Si/m-ZnO NSs substrates of R6G at 10^−2^ M concentration. The obvious Raman enhancement of Si/Ag NPs can be observed to R6G at 10^−7^ M concentration. At the same concentration of R6G, the intensities of the SERS signal of Si/p-ZnO NSs/Ag NPs and Si/m-ZnO NSs/Ag NPs substrates are much stronger than that of Si/Ag NPs substrates. This means that the presence of p-ZnO NSs or m-ZnO NSs can amplify the electromagnetic field enhancement effect. To verify the mechanism, the local electromagnetic field distributions of different structures were simulated by the COMSOL software. The simulated structures are Si/p-ZnO NSs, Si/m-ZnO NSs, Si/Ag NPs, Si/p-ZnO NSs/Ag NPs, or Si/m-ZnO NSs/Ag NPs. The thickness of the Si substrate is 1 μm. The diameter of Ag NPs and the gap between Ag NPs for Si/Ag NPs structures are 36 nm and 8 nm, respectively. The diameter of large ZnO NSs, the diameter of small ZnO NSs, the diameter of Ag NPs, and the gap between Ag NPs for Si/p-ZnO NSs/Ag NPs structure are 276 nm, 135 nm, 32 nm, and 3 nm, respectively. The long diameter and short diameter for the m-ZnO NSs are 162 nm and 132 nm, respectively. The diameter of Ag NPs, and the gap between Ag NPs for Si/m-ZnO NSs/Ag NPs structure are 25 nm and 10 nm, respectively. In this simulation, the incident light (532 nm) irradiates perpendicularly onto the substrate surface. The simulated electromagnetic field intensity distributions are shown in Figure 7a–e, and the best 1000 points of corresponding intensities are shown in Figure 7f. For the Si/p-ZnO NSs and Si/m-ZnO NSs substrates, the maximum electromagnetic field intensities (*E_max_s*) are 3.2 times and 4.6 times incident electromagnetic field (*E*_0_), which shows that the electromagnetic field is enhanced due to the presence of the ZnO NSs structures. The enhancement of the electromagnetic field of ZnO NSs is mainly concentrated in the cavity formed by the gap between adjacent ZnO NSs and the gap between ZnO NSs and substrates. Therefore, weak Raman intensities of R6G (10^−2^ M concentration) can be obtained for the Si/p-ZnO NSs or Si/m-ZnO NSs substrates. The *E_max_* of Si/Ag NPs substrate is 7.4 times *E*_0_, which should be ascribed to the LSPR effect in the hot spots that exist at the nanogaps between adjacent Ag NPs [48]. The *E_max_s* of Si/p-ZnO NSs/Ag NPs and Si/m-ZnO NSs/Ag NPs substrates are 59.9 times and 19.2 times *E*_0_, which are higher than those of the Si/Ag NPs substrates. The enhanced electromagnetic intensities are more obvious in the cavities formed by the gap between adjacent ZnO NSs and the gap between ZnO NSs and substrates. Compared with the Si/m-ZnO NSs/Ag NPs substrate, the Si/p-ZnO NSs/Ag NPs substrate has stronger electromagnetic field enhancement. The enhanced electromagnetic intensities of the two core-satellite nanostructures can be attributed to the more and stronger hot spots due to the following factors: (i) the higher density of Ag NPs on the three-dimensional ZnO NSs due to the increased surface area; (ii) the effectively regulated shape, size and gap of Ag NPs due to the rough surface of ZnO NSs from the abundant anisotropic nanostructures; and (iii) the formed optical cavities from the gap between adjacent ZnO NSs and the gap between ZnO NSs and substrates.

## 3. Conclusions

In summary, the two core-satellite nanostructures of metal oxide/metal nanoparticles were successfully applied to the SERS substrate assembling monodispersed small Ag NPs on large p-ZnO NSs or m-ZnO NSs core. An ultralow LOD of 1 × 10^−13^ M is achieved in the two core-satellite nanostructures with R6G as the probe molecule. Compared with the Si/Ag NPs substrate, the two core-satellite nanostructures of Si/p-ZnO NSs/Ag NPs and Si/m-ZnO NSs/Ag NPs SERS substrates have higher EF of 2.6 × 10^8^ and 2.5 × 10^8^ for R6G as the probe molecule due to the enhanced electromagnetic field. The two core-satellite nanostructures of p-ZnO NSs/Ag NPs and m-ZnO NSs/Ag NPs have great application potential in the low-cost massive production of large-area SERS substrates due to their excellent SERS effect and simple preparation process without any template.

## Figures and Tables

**Figure 1 nanomaterials-12-01286-f001:**
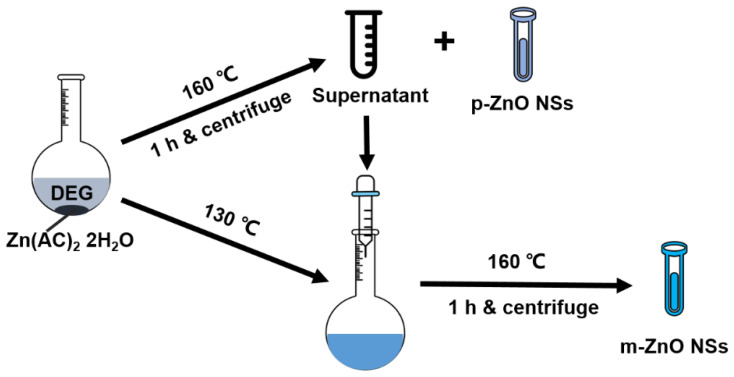
Schematic diagram for the fabrication process of p-ZnO NSs and m-ZnO NSs by the pyrolysis method.

**Figure 2 nanomaterials-12-01286-f002:**
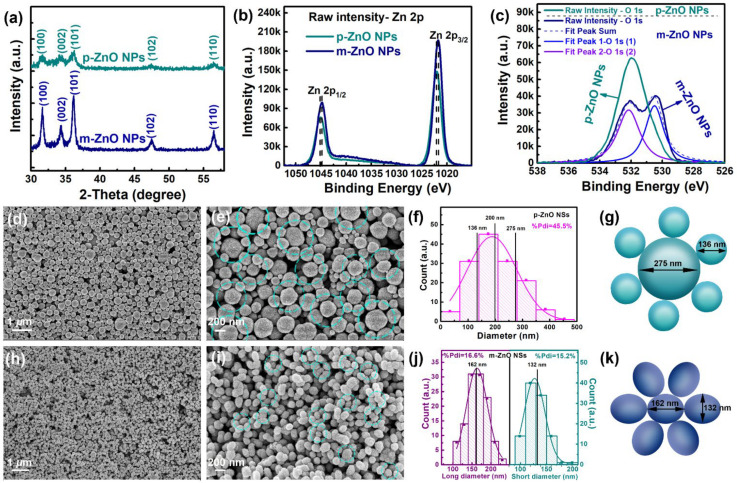
XRD spectra (**a**) and XPS spectra of (**b**) Zn 2p and (**c**) O 1s electron regions for the p-ZnO NSs and the m-ZnO NSs. SEM images of p-ZnO NSs (**d**,**e**) and m-ZnO NSs (**h**,**i**). Statistical histograms for the diameters of the p-ZnO NSs (**f**) and the m-ZnO NSs (**j**). Schematic diagram of p-ZnO NSs (**g**) and the m-ZnO NSs (**k**).

**Figure 3 nanomaterials-12-01286-f003:**
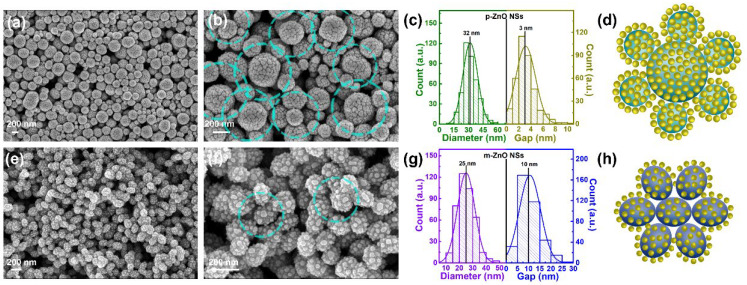
SEM images of p-ZnO NSs/Ag NPs (**a**,**b**) and m-ZnO NSs/Ag NPs (**e**,**f**). Statistical values for the diameters of Ag NPs and gaps between adjacent Ag NPs on the surface of the p-ZnO NSs (**c**) and the m-ZnO NSs (**g**). Schematic diagram of p-ZnO NSs/Ag NPs (**d**) and m-ZnO NSs/Ag NPs (**h**).

**Figure 4 nanomaterials-12-01286-f004:**
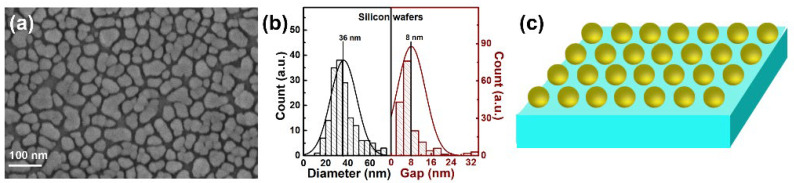
SEM images (**a**), statistical histograms for the diameters (**b**), and schematic diagram of the Ag NPs on the surface of Si substrate (**c**).

**Figure 5 nanomaterials-12-01286-f005:**
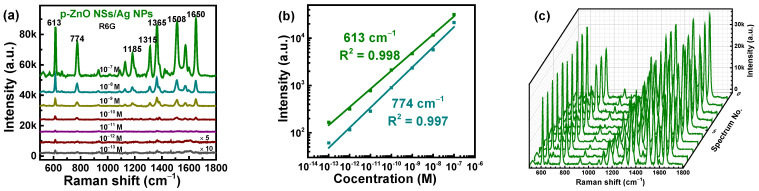
Raman spectra of R6G with different concentrations on the Si/p-ZnO NSs/Ag NPs (**a**) and Si/m-ZnO NSs/Ag NPs (**d**) substrates. Calibration curve of Raman intensity at 613 cm^−1^ versus the concentration of R6G (**b**) and (**e**). Ten Raman spectra of R6G with a concentration of 10^−7^ M on the Si/p-ZnO NSs/Ag NPs substrate from different batches (**c**). Intensities distribution of 613 cm^−1^ peaks of the ten Raman spectra for the Si/p-ZnO NSs/Ag NPs or Si/m-ZnO NSs/Ag NPs substrates (**f**).

**Figure 6 nanomaterials-12-01286-f006:**
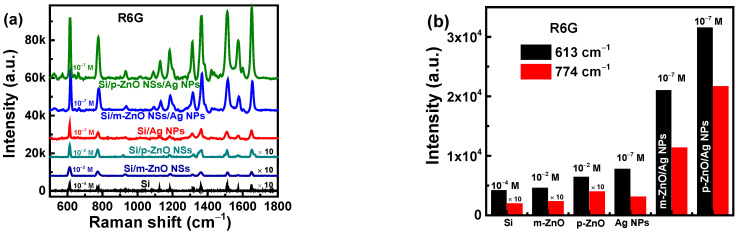
(**a**) Raman spectra of R6G with concentrations of 10^−7^ M, 10^−4^ M, or 10^−2^ M for different substrates. (**b**) Corresponding Raman intensities of the 613 cm^−1^ and 774 cm^−1^ peaks for different substrates.

**Figure 7 nanomaterials-12-01286-f007:**
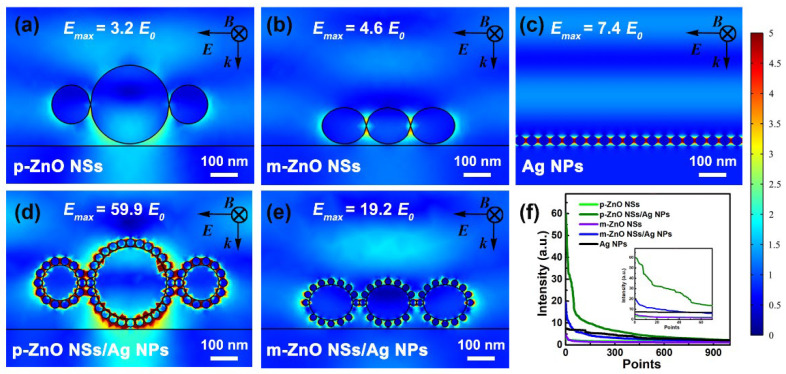
The x-z views of simulated electromagnetic field distributions at 532 nm for the Si/p-ZnO NS (**a**), Si/m-ZnO NS (**b**), Si/Ag NP (**c**), Si/p-ZnO NS/Ag NP (**d**), and Si/m-ZnO NS/Ag NP (**e**) substrates. Corresponding simulated electromagnetic field intensities of best 1000 points for different substrates (**f**).

**Table 1 nanomaterials-12-01286-t001:** Raman performance of recent typical SERS substrates based on the combination of ZnO micro/nanostructures and noble metal nanoparticles (RhB^1^ is Rhodamine B; 4-ATP^2^ is 4-Aminothiophenol).

Structures	Template-Free Synthesis	Probe Molecules	EF	LOD	Ref.
ZnO nanocone/Au NPs/Ag nanoclusters	No	RhB^1^	6.5 × 10^9^	10^−12^ M	[30]
Porous ZnO nanosheets/Ag NPs	No	R6G	3.5 × 10^7^	10^−11^ M	[31]
Urchin-like ZnO-nanorod/Ag NPs	No	4-ATP^2^	1.7 × 10^7^	10^−12^ M	[32]
ZnO nanorods/Ag NPs	Yes	R6G	1.9 × 10^7^	10^−10^ M	[33]
p-ZnO NSs/Ag NPs	Yes	R6G	2.6 × 10^8^	10^−13^ M	This work
m-ZnO NSs/Ag NPs	Yes	R6G	2.5 × 10^8^	10^−13^ M	

## Data Availability

Not applicable.

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
