# Peer review of "High-Performance Surface-Enhanced Raman Scattering Substrates Based on the ZnO/Ag Core-Satellite Nanostructures"

_nanomaterials, 2022, doi:10.3390/nano12081286_

Round 1

Reviewer 1 Report

The manuscript by Q. Sun et al. reports on an interesting topic about ultra-low limit detection of Rhodamine 6G by ZnO/Ag nanostructures as SERS templates. The deposition process of the ZnO nanostructures was performed via a chemical pyrolysis method without any template while the Ag NPs were prepared by thermal evaporation without any postdeposition annealing. The authors claim significant enhancement in the active surface and the electromagnetic field resulting in low detection limit of molecules. However, there are critical points that should be addressed and the authors should proceed to major revisions before the publication of this manuscript.

(1) The authors followed a pyrolysis method with a two-stage reaction process to fabricate the p-ZnO and the m-ZnO nanospheres (NSs) patterns. They claim that both NSs present a core-satellite morphology while from the SEM images (Fig 2a,b and Fig 3a,b) it is obvious that the p-ZnO NSs show larger NSs with wide bimodal size distribution, and the m-ZnO NSs show much smaller NSs with more unimodal size distribution. The reason for this analysis is unclear as there is no correlation between these particular features (large/small sphere diameter and long/short sphere diameter) and the Ag NPs, for which only the average size and gap are mentioned. Furthermore, the authors claim the formation of core-satellite structures. What are the forces that hold the particles together and form these nanostructures?

(2) During the last 2 decades, SERS and the field enhancement of noble metals have been studied widely and in-depth. There is a lack of important literature in the manuscript for SERS and the methods as well as the comparison with established and well-known fabrication processes which can have comparable detection limits and results like Nanosphere Lithography [Haynes CL, Van Duyne RP J Phys Chem B. 105, 5599 (2001)], Electron Beam Lithography [Haynes CL, et al., J Phys Chem B. 107, 7337 (2003)], Laser Annealing [Beliatis MJ, et al., Opt Lett 36 1362 (2011)]. More references are required.

(3) The authors present the LOD and the enhancement factor of the Ag decorated ZnO nanostructures in R6G dye. The Ag NPs size and distribution of Ag/ZnO NSs and Ag reference NPs should be correlated with corresponding Ag NPs from the literature that present similar size and optical properties in order to determine the efficiency of the Ag/ZnO nanostructures as SERS templates.

(4) Page 4, lines 142-144. The authors claim that “the p-ZnO NSs and m-ZnO NSs provide supportive structures for the deposition of Ag NPs on them, which play a key role in controlling the size and shape of Ag NPs”. More details and connections with the literature should be provided.

(5) Ag growth typically exhibits a pronounced and uncontrolled 3D morphology on weakly interacting substrates like ZnO, SiO2 [Campbell, C. T. J. Chem. Soc., Faraday Trans., 92 (9) 1435 (1996), Zhang et al., J. Phys. D: Appl. Phys. 42, 065303 (2009)]. The authors should link this behavior with their results in order to make clear the growth mechanism that affects the development of Ag NPs on ZnO NSs. Furthermore, as the crystallinity of ZnO NSs may affect significantly the growth morphology of the Ag NPs, the authors should provide information about the crystallinity and the grain size of the ZnO NSs.

(6) In the manuscript there is no reference on the quality of the ZnO. It is recommended that the XPS data be provided to verify the quality of the ZnO NSs and determine how the two-stage pyrolysis process affects the process.

(7) The authors performed only deposition of 6 nm Ag via thermal evaporation. What is the effect of the thickness/quantity of Ag on the size and morphology of the NPs on the ZnO NSs?

(8) According to the literature [F. De Angelis, et al., Nat. Photonics 5 (2011) 682–687, J.M. Rui Tan, et al., Phys. Chem. Chem. Phys. 16 (2014) 26983–26990, etc.], hydrophobicity can affect the dewetting properties during the drying process resulting in a significant increase in the concentration of the R6G and the SERS signal. The authors report that at concentrations lower than 10-14 M, the SERS signal can be only obtained from the edge of the evaporation imprint due to the higher agglomeration of R6G dye. This phenomenon is called the coffee stain effect [R.D. Deegan, et al, Nature 389 (1997) 827–829]. The hydrophobic properties of the Ag/ZnO nanostructures should be reported and correlated with the LOD and the enhancement.

(9) In Figure 6b, the comparison of the intensity values for the different substrates is misleading. The intensity of the 613 nm peak appears with a gradual increase from Si substrate to Ag NPs template but the R6G concentration is not the same. According to the figure, the peak intensity of flat Si substrate should have been higher than m-ZnO and p-ZnO, if the same concentration was used, which means that the flat Si has a higher detection. The authors should explain this behavior.

(10) The authors show that the Ag/p-ZnO NSs reveal a stronger electromagnetic enhancement resulting in lower LOD and higher enhancement factor although Ag NPs on p-ZnO NSs present a much worse distribution compared to m-ZnO NSs. They claim that p-ZnO NSs can provide a more favorable growth environment for the Ag NPs to improve the LSPR effect. The authors should explain this behavior extensively from the literature indicating the effect of the ZnO templates on the optical/plasmonic properties of the Ag NPs.

(11) Page 8. The numbering in line 282 should be (iii) instead of (ii) and in Figure 7 caption, line 288 is interrupted and placed in line 284.

(12) Literature that authors should consider: i) K. Kneipp, et al., Phys. Rev. Lett. 78 (1997) 1667–1670, ii) D.L. Jeanmaire, R.P. Van Duyne, J. Electroanal. Chem. 84 (1977) 1–20, iii) M.F. Cardinal, et al., Chem. Soc. Rev. 46 (2017) 3886–3903, iv) P.L. Stiles, et al., Annu. Rev. Anal. Chem. 1 (2008) 601–626, v) B. Sharma, et al., Mater. Today 15, 16 (2012), vi) R.J.C. Brown and M.J.T. Milton, J. Raman Spectrosc. 39, 1313 (2008), vii) J. Langer, et al., ACS Nano 14, 28 (2020).

Author Response

Thank you very much for your valuable comments and suggestions, which are very useful for us to further improve the article quality. We have made careful corrections according to your suggestions. The detailed reply is in the attachment. Thank you very much for your support.

Author Response

(The authors gave the same response as above.)
